Uncovering tissue-specific endophytic microbiota composition and activity in Rhizophora mangle L.: a metagenomic and metatranscriptomic approach

Cárdenas-Hernández Valentina valentina.cardenas.hernandez@correounivalle.edu.co
Lemos-Lucumi Cesar
Toro-Perea Nelson
Department of Biology, Universidad del Valle , Cali , Valle del Cauca , Colombia
Phitsuwan Paripok
Electronic publication date: 2025 Aug 28
Publication date: 2025
Volume: 13
Electronic Location ID: e19728
Received 2025 Feb 18; Accepted 2025 Jun 18
Copyright: ©2025 Cárdenas-Hernández et al.
Copyright year: 2025
Copyright holder: Cárdenas-Hernández et al.
License: This is an open access article distributed under the terms of the Creative Commons Attribution License, which permits unrestricted use, distribution, reproduction and adaptation in any medium and for any purpose provided that it is properly attributed. For attribution, the original author(s), title, publication source (PeerJ) and either DOI or URL of the article must be cited.
License URL: https://creativecommons.org/licenses/by/4.0/

Keywords: Differential expression, Endophytic microbiome, Functional annotation, Plant-endophyte interaction, Rhizophoraceae, RNAseq, WMS

Funding: The Science Fund Technology and Innovation of the General Royalty System with BPIN: 2021000100488 The Universidad del Valle with CI:71323 This research was funded by the Science Fund Technology and Innovation of the General Royalty System with BPIN: 2021000100488 and by the Universidad del Valle with CI:71323. The funders had no role in study design, data collection and analysis, decision to publish, or preparation of the manuscript.

==============================
The interaction of mangrove trees with endophytic microorganisms contributes to the successful establishment of these plants in the challenging intertidal environment. The red mangrove, Rhizophora mangle L. (Rhizophoraceae), is one of the dominant species in mangrove ecosystems and is characterized by the provision of several ecologically relevant services. In this work, we integrated metagenomics and metatranscriptomics to perform a robust characterization of the community of endophytic microorganisms associated with R. mangle leaf and root tissues. The microbiota were characterized at taxonomic and functional levels, and abundance and gene expression profiles were compared between these two plant tissues. We found that the endophyte community consisted mainly of bacteria and eukaryotes, which were the most active groups at the transcriptional level, while archaea and viral groups were identified in lower abundance and expression. In addition, the results show that the community of endophytic microorganisms changes depending on the tissue type, with root-associated microorganisms being the most abundant at the metagenome level and active at the metatranscriptome level. It was also found that R. mangle endophytes actively contribute to key functions for adaptation to an intertidal ecosystem with high human intervention, such as salinity tolerance and degradation of heavy metals and xenobiotic compounds. Thus, according to the functions found and contributed by the endophyte community of red mangrove leaf and root tissues, it can be concluded that these microbial communities are crucial for the survival of R. mangle in the extreme environment of mangrove forests. This study provides a solid basis for future research aimed at understanding the role of plant-endophyte interactions.

Introduction

The species Rhizophora mangle L., known as the red mangrove, belongs to the family Rhizophoraceae and is one of the dominant tree species in the intertidal vegetation of tropical and subtropical regions (DeYoe et al., 2020). Among the various ecosystem services this species provides are coastal protection, carbon sequestration, and food supply (Hamza et al., 2024). In addition, R. mangle has traditionally been used for the extraction of wood and fuel resources throughout its distribution (Friess, 2016; Huxham et al., 2017). This species exhibits various morphological, physiological, and molecular adaptations that allow it to thrive in the extreme conditions of mangrove ecosystems (Srikanth, Lum & Chen, 2016). For example, it develops specialized stilt roots that provide stability in unstable soils and protect trees from tidal action (Srikanth, Lum & Chen, 2016). The leaves of Rhizophora species, in turn, contain mucilaginous cells that help retain water, influence ion transport, and contribute to salinity tolerance (Naskar, Mondal & Ankure, 2021). In addition to these plant-specific adaptations, interactions with endophytic microorganisms also play a key role in the successful establishment of mangrove trees in the stressful environment of the intertidal zone (Deepika, Lavanya & Sridevi, 2023). Subedi et al. (2022) demonstrated that inoculation of R. mangle seedlings with endophytes increased their resistance to salinity.

Endophytic microorganisms provide multiple benefits to their host plants, such as protection against biotic and abiotic stresses, growth promotion, and physiological enhancement, all through the production of diverse bioactive compounds (Nandhini et al., 2020). In the case of mangrove endophytes, they have attracted significant biotechnological interest, as they have been identified as a reservoir of these compounds (Chatterjee & Abraham, 2020). Notable examples include phenols, alkaloids, quinones, steroids, terpenoids, polysaccharides, proteins, and enzymes (Bolivar-Anillo et al., 2023). For instance, endophytes isolated from the mangrove species Acanthus ilicifolius L. have been shown to produce indole-3-acetic acid (IAA) and solubilize phosphates—compounds that enhance host growth under saline conditions (Deepika, Lavanya & Sridevi, 2023). Several studies have also highlighted the biotechnological potential of these microorganisms and their metabolites, which exhibit antimicrobial, antimutagenic, and antioxidant activities (Hamzah et al., 2020; Sopalun et al., 2021).

Given the importance of plant–endophyte interactions, several researchers have sought to characterize the endophytic microorganisms associated with leaf and root tissues of R. mangle, initially using culture-based approaches to isolate and identify fungal species present in these tissues (Ananda & Sridhar, 2002; Costa, Maia & Cavalcanti, 2012; Sital, Da Silva & Pestano, 2020). More recently, the development of next-generation sequencing technologies has greatly improved the identification of fungal and bacterial endophytes through the analysis of ITS and 16S rRNA sequences, respectively (Da Silveira Bastos et al., 2024). However, despite these methodological advances, Harrison & Griffin (2020) report that mangrove ecosystems remain among the least studied in terms of endophyte diversity. Most of the available information still focuses on cultivable fungal groups, creating a significant knowledge gap. In this context, robust molecular tools are needed to expand the identification of unculturable endophytic species and to shed light on the ecological roles these microorganisms play. Kaul, Sharma & Dhar (2016) emphasize that integrated approaches, such as metagenomics and metatranscriptomics, offer a more comprehensive understanding of plant–microbe relationships by providing information on both microbial identity and gene expression.

The study of endophytic microorganisms in mangrove forests presents several challenges. These ecosystems harbor a high diversity of bacteria, archaea, fungi, and viruses, whose interactions are highly complex and influenced by spatiotemporal environmental variability. Furthermore, due to the unique and fluctuating conditions of mangroves, a large portion of this microbial diversity remains uncultured and poorly characterized (Nagarajan et al., 2025; Selvaraj et al., 2025). In recent years, metabarcoding and metagenomic approaches have been increasingly used to explore the microbial communities associated with mangrove plants (Yao et al., 2020; Zhuang et al., 2020; Sui et al., 2023). For instance, Yao et al. (2020) characterized the epiphytic and endophytic bacterial communities of six mangrove species by sequencing the V4 hypervariable region of the 16S rRNA gene. Sui et al. (2023) targeted the V3–V4 region of the same gene to investigate the root endosphere, rhizosphere, and bulk soil associated with four mangrove species. Zhuang et al. (2020) combined metabarcoding and metagenomic strategies to generate a taxonomic and functional profile of microbial communities inhabiting the non-rhizosphere, rhizosphere, episphere, and endosphere of Kandelia obovata Sheue, H.Y. Liu & J.W.H. Yong. In the case of R. mangle, most studies have focused primarily on bacterial communities (Scherer, Mason & Mast, 2022), and limited information is available regarding the taxonomic and functional profiles of its endophytic microbiota. Despite growing interest in R. mangle-associated microbiomes, knowledge of the structure and function of these microbial communities remains scarce. In this context, the integration of metagenomic and metatranscriptomic approaches represents a powerful strategy to overcome the limitations of individual methodologies, enabling a more holistic characterization of non-culturable endophytic taxa, their genomes, and their gene expression profiles (Kaul, Sharma & Dhar, 2016; Saenz et al., 2022).

Given the ecological and economic importance of R. mangle endophytes, this research aims to integrate metagenomic and metatranscriptomic approaches to achieve a robust characterization of the endophytic microbiota present in the leaf and root tissues of R. mangle. Beyond addressing the knowledge gap surrounding non-culturable microbial diversity, this study contributes to a more comprehensive understanding of plant–endophyte interactions from a multidisciplinary perspective. Additionally, the insights gained may support future biotechnological applications and conservation strategies in mangrove ecosystems.

Materials & Methods

Experimental design and sample collection

The samples were collected from a mangrove area located in the Bay of Buenaventura, southwestern Colombia (Fig. S1). This mangrove forest spans 1,984.5 ha and is heavily impacted by human activity, including pollution from garbage and heavy metals (SIDAP (Sistema Departamental de Áreas Protegidas, CVC), 2011; Gamboa-Garcia et al., 2020; Bolivar-Anillo et al., 2023). Sampling took place on June 14, 2023, at the site where the Dagua River meets the Pacific Ocean (3°52′44.887″N, 77°3′44.577″W). Meteorological and tidal conditions on the sampling day are provided in Table S1 and Fig. S2. Sampling occurred between 8:00 and 11:00, coinciding with the lowest tide (Fig. S2). The collection followed the experimental design outlined in Fig. S3. Leaf and root tissue samples were taken from three R. mangle individuals, spaced approximately 10 m apart. The collection was conducted under the “Framework Permit for the Collection of Specimens of Wild Species of Biological Diversity for Non-Commercial Scientific Research” issued to the Universidad del Valle by the National Environmental Licensing Agency (ANLA) through Resolution 1070 on August 28, 2015. Before collecting, the root and leaf surfaces were cleaned according to a modified version of the protocol by Correa-Galeote, Bedmar & Arone (2018). Specifically, 2% sodium hypochlorite was applied for 10 min, followed by a 5-minute treatment with 70% ethanol, and a final wash with distilled water for 3 min. The protocol was adjusted because it was not feasible to immerse the mangrove roots in ethanol, so the exposure time to the reagents was extended to ensure thorough elimination of rhizospheric microorganisms. For leaf tissue, only healthy leaves without signs of mechanical damage or disease were collected directly from the trees. For root tissue, a section of the root exposed to tidal action was selected, and a sterile drill was used to extract only the internal tissue. All collected tissues were immediately stored in liquid nitrogen and transported to the Molecular Biology Laboratory of Universidad del Valle, where they were stored at −80 °C until processing. To characterize the physicochemical conditions of the sampling area, a soil sample was collected and analyzed by AGRILAB laboratory (Bogotá, Colombia), while a water sample was analyzed using the HANNA HI9829 multiparameter meter (Table S2).

DNA extraction and molecular identification of individuals

The collected tissues, both leaves and roots, were macerated with liquid nitrogen and the subsequent DNA extraction was performed with the “InviSorb® Spin Plant Mini Kit” (Invitek, Hayward, CA, USA), following the protocol recommended by the manufacturer, including, as a modification, a second maceration using a DREMEL 3000 with a sterilized pistil, after the addition of the lysis buffer. The DNA obtained was quantified by fluorometry using the Qubit 4 (Thermo Fisher Scientific, Waltham, MA, USA), using the dsDNA BR Assay Kits (Thermo Fisher Scientific, Waltham, MA, USA). In addition, 0.8% agarose gel electrophoresis was performed to evaluate the integrity of the extracted genomic DNA.

To confirm the identity of the mangrove trees, the atpI-atpH and psbJ-petA systems, two non-coding regions of the chloroplast, were amplified and sequenced. The primers used and the thermal profile were as described by Ceron-Souza et al. (2012). For Sanger sequencing, the amplicons were sent to Macrogen in South Korea. The sequences were then mapped against the NCBI database to confirm that the collected individuals belonged to the species R. mangle.

RNA extraction

The same tissue samples from the three replicates of leaves and roots used for DNA extraction were used for RNA extraction. We used 250 mg of tissue and followed the protocol described by White and coworkers (2008), based on the use of CTAB detergent, with some modifications. The protocol was modified to incubate the sample overnight in a volume of isopropanol at −20 °C instead of using 8M lithium chloride and incubating at 4 °C. The extracted RNA was quantified using the RNA BR Assay Kits (Thermo Fisher Scientific, Waltham, MA, USA) on a Qubit 4 (Thermo Fisher Scientific, Waltham, MA, USA). To assess the integrity of the extracted total RNA, 1.5% agarose gel electrophoresis was performed.

Metagenome sequencing

Once it was confirmed that the sampled individuals belonged to the R. mangle species, the extracted DNA was sent to Macrogen in South Korea for shotgun sequencing of the metagenomes. The samples were shipped with a minimum of 1.0 µg of DNA. For library construction, the TruSeq DNA nano kit was used. The NovaSeq6000 platform (Illumina, San Diego, CA, USA) was then used for paired-end sequencing of 150 bp fragments, generating 20 GB of information for each sample. This provided DNA sequences of the three R. mangle trees and the DNA of the community of endophytic microorganisms associated with their leaf and root tissues.

Metatranscriptome sequencing

Total RNA was also sent to Macrogen in South Korea, where the RNAseq process was performed. Samples were sent with a minimum RNA content of 0.5 µg and were prepared in a solution of sodium acetate (3M) and ethanol according to Macrogen’s recommendations. For library construction, the TruSeq stranded total RNA kit, and the Ribo-Zero Gold kit were used to remove ribosomal RNA. The NovaSeq6000 platform (Illumina, San Diego, CA, USA) was used to perform paired-end sequencing of 100 bp fragments, generating 40 million reads for each sample. In this way, the mRNA sequences of R. mangle and the community of endophytic microorganisms associated with leaf and root tissues were recovered. Replicate three of root total RNA showed an excess of polysaccharides and phenolics, multiple extraction and clean-up attempts were made to remove these molecules, but it was not possible to obtain a positive result. Therefore, the library for root tissue total RNA replicate number three (RmR3) could not be constructed and was not included in the following analyses.

Bioinformatic analysis of metagenomes

First, the raw reads obtained from the shotgun sequencing were processed. Using the Trimmomatic v 0.39 program, sequences with a Phred score lower than 20, which were considered of poor quality, were eliminated (Bolger, Lohse & Usadel, 2014). Likewise, reads or sequences corresponding to the DNA of R. mangle were eliminated, for which, the genome of the species, (CA Lemos-Lucumí, 2025, unpublished data), was used as a reference. This subtraction was performed using Bowtie2 v2.5.4 and SamTools v1.21 (Li et al., 2009; Langmead & Salzberg, 2012). Only those reads that did not align with the R. mangle genome were recovered, thus obtaining the sequences of the endophytic microorganisms.

Once the clean and exclusive reads of the microorganisms were obtained, the taxonomic and functional analysis was performed. First, Kraken2 v2.1.3 software was used to perform taxonomic assignment, with the default classification parameters (Wood, Lu & Langmead, 2019). For this process, the NCBI databases of Bacteria, Fungi, Protozoa, Archaea, and Viruses were used (Wood, Lu & Langmead, 2019). The unclassified reads were not used for taxonomic diversity analyses; however, they were used for metagenome assembly and subsequent functional annotation analyses. Subsequently, the clean reads of all samples were used to perform the assembly of the endophytic metagenome of R. mangle using the program MetaSPAdes v4.0.0 (Nurk et al., 2017), the assembly was performed for leaves and roots separately. The quality of assemblages was assessed with MetaQUAST v5.3 (Mikheenko, Saveliev & Gurevich, 2016). In addition, using CD-Hit v4.8.1, with 95% similarity and 90% coverage, redundant sequences were removed from the metagenome (Fu et al., 2012). Regarding functional analysis, DIAMOND v.2.1.9 software was used to annotate the metagenome against the NCBI Refseq database of Archaea, Bacteria, Fungi, Protozoa and Viruses (O’Leary et al., 2016). Also, the program Prokka v1.14.5 (Seemann, 2014), was used to extract the coding sequences (CDS) of the metagenomes. Subsequently, using eggNOG-mapper v2.1.12 (Cantalapiedra et al., 2021) the obtained CDS were annotated against the Kyoto Encyclopedia of Genes and Genomes (KEGG) and the Orthologs Groups (COG) databases, (Kanehisa, 2002; Tatusov et al., 2003). The functional annotation results were analyzed through the MicrobiomeProfiler package of R (RStudio Team, 2020; Chen & Yu, 2023). This was done with the aim of identifying the possible metabolic categories where the different coding sequences specific to metagenomes can be found.

Bioinformatic analysis of metatranscriptomes

Raw RNA reads were processed following the steps described above for DNA reads to remove poor quality sequences and reads corresponding to the host plant. In addition, reads corresponding to rRNA were removed using the SortMeRNA v4.3.7 software (Kopylova, Noe & Touzet, 2012) and the Silva ribosomal RNA databases (Quast et al., 2012). Once the mRNAs unique to the endophytic microorganisms were obtained, metatranscriptome assembly, functional annotation, and differential expression and functional enrichment analyses were performed. The statistical power of this experimental design, calculated using the RNASeqPower package (Hart et al., 2013), is 0.998 (Table S3). This calculation was performed with a sequencing depth of 70X and a target effect size of 2.

For metatranscriptome assembly, the program rnaSPAdes v4.0.0 (Bushmanova et al., 2019) was used with default settings, using both leaf and root samples. Then, the metatranscriptome coverage was improved by using the assembled metagenomes, following the “covered cds” script of the MetaGT workflow v0.1.0 (Shafranskaya et al., 2022). For functional annotation, protein sequences of Bacteria, Fungi, Protozoa, archaea, and viruses from the NCBI RefSeq database were used and metatranscriptome annotation was performed using the DIAMOND v.2.1.9 program (O’Leary et al., 2016; Buchfink, Reuter & Drost, 2021). The resulting annotations were converted into tables of counts using the “DIAMOND analysis counter” script of the Samsa2 v2.2.01 workflow (Westreich et al., 2018). These counts were used as input for differential expression analyses using the DESeq2 v 3.20 package of the R program (Love, Huber & Anders, 2014; RStudio Team, 2020), with only those annotations containing more than three counts summed across all samples included. Differential expression per tissue was evaluated, and gene expressions with an adjusted p-value less than 0.05 and a log2FoldChange greater than two or less than minus two were considered significant. Finally, for functional enrichment analysis, genes that showed significant differential expression were annotated against the KEGG (Kanehisa, 2002) and EggNOG (Jensen et al., 2007) databases using the eggNOG-mapper v2.1.12 program (Cantalapiedra et al., 2021). The results were analyzed using the MicrobiomeProfiler package in R (RStudio Team, 2020; Chen & Yu, 2023).

Comparative functional bioinformatics analysis between metagenome and metatranscriptome

To further investigate the plant-endophyte interaction, the PGPg_finder v1.1.0 workflow (Pellegrinetti et al., 2024) was used to annotate DNA and RNA samples against the PLaBAse database, which contains unique information on genes relevant to this interaction (Patz et al., 2021). This integrated analysis allowed us to observe both the presence of genes in the metagenome and their expression in the metatranscriptome, demonstrating how genetic and transcriptomic profiles complement each other. This facilitated the determination of the contribution of endophytic microorganisms in plant metabolism, degradation of xenobiotic compounds, tolerance to salinity and resistance to biotic, abiotic and heavy metal stresses.

Statistical analysis

The files resulting from the taxonomic annotation performed with Kraken2 v2.1.3 for the DNA and RNA libraries were transformed into “.biom” files using the kraken-biom v1.0.1 program (Dabdoub, 2016) and imported into Rstudio as an object of the Phyloseq v 3.20 package (McMurdie & Holmes, 2013; RStudio Team, 2020). A filter was performed according to the number of reads per species, in the case of DNA samples those species with an abundance of less than 10 reads were removed. Subsequently, alpha diversity was estimated by calculating the observed richness and the Shannon, Simpson and Chao1 indices for DNA and RNA libraries (McMurdie & Holmes, 2013). For beta diversity, principal coordinate analysis (PCoA) was performed based on Bray-Curtis and Jaccard distance matrices (McMurdie & Holmes, 2013). For RNA samples, a principal component analysis (PCA) based on NCBI gene annotation was performed using the DESeq2 v 3.20 package in R to determine whether there were differences in gene expression in the microorganism community according to the tissue with which it was associated.

Results

The aim of this research was to combine metagenomics and metatranscriptomics to provide a taxonomically and functionally robust characterization of the endophytic microorganisms associated with the leaf and root tissues of R. mangle. To address this objective, we performed shotgun sequencing and RNAseq on surface-sterilized leaf and root tissues.

Shotgun sequencing of the six DNA samples generated 481,358,005 raw reads, of which 54,391,543 were retained after processing. Sequencing of the five RNA samples generated 405,134,926 raw reads, of which 5,466,396 clean reads were retained (Table S3). The processed reads of both DNA and RNA are available in the NCBI database (Bioproject number: PRJNA1215819). This BioProject contains sequencing data from a broader study on the endophytic microbiota of various mangrove species. However, this article focuses exclusively on the 11 samples derived from R. mangle (six DNA and five RNA samples). These specific samples were generated, analyzed, and interpreted in the context of this work. The corresponding BioSample accession numbers are: SAMN46420086 to SAMN46420096.

The high percentage of eliminated reads can be explained by the fact that the samples were derived from R. mangle tissue, so most of the extracted nucleic acids corresponded to the host genome and transcriptome. The retained reads represent the endophytic microorganisms associated with the roots and leaves of R. mangle. This explains why the sequencing data available in the SRA is substantially smaller in size than the total volume initially generated.

Taxonomic profile of the endophytic microbiome of R. mangle leaves and roots

The taxonomic analysis of the 12 DNA libraries allowed the classification of between 8.19 and 24.7% of the reads. It was observed that the endophytic microbiome of leaf and root tissues consisted mainly of bacteria, followed by eukaryotes, and in lower abundance archaea and viruses (Fig. 1A). The most abundant phyla in leaves and roots were Pseudomonadota and Actinomycetota. While Bacillota, Bacteroidota and Apicomplexa were abundant in leaf tissue samples (Fig. 1B). Already at the family level, the most abundant taxa were Streptomycetaceae and Nitrobacteriacea in both tissue types. The families Lactobacillaceae, Nocardioidaceae, Mycobacteriaceae and Pseudomonadaceae were also identified as abundant in some of the samples (Fig. 1C). The most abundant genera in both leaves and roots were Bradyrhizobium and Streptomyces. Also, the genera Lactiplantibacillus, Mycobacterium, Nocardioides, and Pseudomonas showed high abundance in some of the leaf tissue samples (Fig. 1D). The detailed taxonomic composition of the endophytic microbiota of R. mangle, for each of the leaf and root replicates, is shown in Figs. S4–S9.

Figure 1 Taxonomic composition of the endophytic microbial community associated with R. mangle leaf and root tissues, shown at four levels: (A) kingdom, (B) phylum, (C) family, and (D) genus.

Each bar represents one of three replicates per tissue type. Sample names follow this format: Rm = R. mangle; L = leaf tissue; R = root tissue.

Regarding the composition of the endophytic microbial community of R. mangle, it was found that it was influenced by the type of plant tissue. This could be demonstrated descriptively, since in the PCoA two groups were formed depending on the tissue (Fig. 2). On the other hand, the observed richness was higher in root tissue samples than in leaf tissue samples (Fig. S10). This pattern was also evident for the Shannon index, with all root tissue samples having the highest values (Fig. S10). For Simpson’s index, all samples had a value of approximately 0.99, except for the second leaf tissue sample (RmL2), which had an index of 0.79 (Fig. S10).

Figure 2 Principal coordinate analysis (PCoA) based on Bray–Curtis dissimilarities, showing the taxonomic composition of the endophytic microbiome in R. mangle as a function of tissue type (leaf vs. root).

Each point represents one of three replicates per tissue. Sample names follow this format: Rm = R. mangle; L = leaf tissue; R = root tissue.

Metagenome and metatranscriptome assembly of the R. mangle endophytic microbial community

The metagenomes by tissue showed differences in assembly parameters (Table 1). The root tissue metagenome was 140.8 times larger than the leaf tissue metagenome with a total length of 915 M and 6.5 M base pairs, respectively (Table 1). In addition, the N50 for the leaf tissue metagenome was larger than that for the root tissue metagenome. For the metatranscriptome, the assembly had a total length of 2.4M bp with an N50 of 711 (Table 1).

Functional potential of the R. mangle endophyte community: functional annotation of metagenomic coding sequences

The annotation of the metagenomes against the EggNOG database showed that for most of the coding sequences (CDS) their putative function is not known (Fig. 3). As for the CDS that could be assigned within a functional category, it was found that for both the leaf and root tissue metagenomes, CDS related to the following categories are abundant: replication, recombination and repair; transcription and translation; ribosomal structure and biogenesis (Fig. 3). Regarding cellular processes and signaling, the categories with the highest abundance of CDS were signal translation mechanisms and cell wall, membrane and envelope biogenesis (Fig. 3). Finally, regarding cellular metabolism, the most relevant category in both tissue types was amino acid transport and metabolism (Fig. 3). Although the metabolic categories are abundant in both tissues, there is a difference in the CDS counts between the two, with the root tissue being the one with the highest counts, with a difference of approximately 100-fold in relation to leaf tissue (Fig. 3). On the other hand, other differences were observed between the metagenomes of leaf and root tissues, with more abundant CDS related to carbohydrate transport and metabolism in leaves (Fig. 3A), and CDS from the category of energy production and conversion in roots (Fig. 3B). Metagenome annotation against the KEGG database, for leaf and root metagenomes, showed that most CDS were assigned to categories related to amino acid, carbohydrate, and nucleotide metabolism, in addition to cell maintenance and signaling functions (Fig. S11).

Table 1 Quality parameters of metagenome and metatranscriptome assemblies from R. mangle leaf and root tissue.

Only one metatranscriptome assembly was performed, see methodology for details.

Assembly parameters	Metagenome	Metatranscriptome	
	Leaf tissue	Root tissue	Leaf + Root tissue	
Number of contigs	3,362	717,990	7,513	
Number of contigs (>= 1,000 pb)	1,127	238,749	147	
Largest contig (bp)	220,657	239,813	5,658	
Total length (bp)	6,566,090	915,928,441	2,430,642	
N50	6,552	1,473	711	
N90	659	594	525	
L50	72	127,705	324	
L90	2,215	549,490	808	

Figure 3 Functional profile of endophytic microorganisms associated with R. mangle leaf and root tissues.

The figure shows the abundance of coding DNA sequences (CDSs) assigned to functional categories in the EggNOG database. (A) Metagenome of leaf tissue. (B) Metagenome of root tissue.

Transcriptional profiling, differential expression and enrichment of functional categories of the metatranscriptome

Annotation against the NCBI database revealed a total of 5,404 active genes present in all analyzed samples. The most highly expressed genes were the NADH dehydrogenase gene, followed by NAD-dependent malic enzyme and ADP ribosylation factor (Table S4). When analyzing the expression by tissue, in addition to the above-mentioned genes, in the endophytic microbiota of leaves glutamine synthetase, catalases and aspartic proteases can be observed in high expression, while in the root community enolase and cytochrome c oxidases transcripts stand out (Table S4). The PCA performed with the 5,404 active genes allowed to determine that there is a difference between the gene expression of the endophytic microbiota of leaves and roots of R. mangle, since it is observed how the samples are completely separated according to the tissue from which they come (Fig. 4). This explains the 81% variability between the first two components.

Figure 4 Principal component analysis (PCA) of functional annotations (RefSeq database) from metatranscriptomic samples of R. mangle.

The analysis includes three replicates of leaf tissue and two replicates of root tissue (see Methodology for details). Sample names follow this format: Rm = R. mangle; L = leaf tissue; R = root tissue.

Differential expression analysis based on RefSeq revealed a total of 44 genes with significant differential expression, of which four were found to be overexpressed in leaves and 39 in roots (Fig. 5A). The genes overexpressed in roots correspond to proteins related to the oxidative phosphorylation cycle, such as cytochrome c oxidase and NADH dehydrogenase, and in leaves to two types of glutamine synthetase (Fig. 5B). Functional enrichment of microbial genes overexpressed in leaves showed amino acid transport and metabolism as the most abundant category in the eggNOG database (Fig. 6A), while enrichment with the KEGG database did not reveal any metabolic pathway with significant abundance. On the other hand, functional enrichment of microbial genes overexpressed in roots revealed cellular processes related to cytoskeleton and energy production and conservation as the most abundant categories according to the eggNOG database (Fig. 6B). In contrast, enrichment using the KEGG database revealed only the oxidative phosphorylation pathway (Fig. S12).

Figure 5 Differential expression of the endophytic microbiome of leaves and roots using the NCBI RefSeq database.

(A) Volcano plot of the 5,404 active genes. Gray: Not significant. Green: Genes with more than 2 or less than -2 in the Log2FoldChange value. Red: Genes the Log2FoldChange criterion and with an adjusted p-value less than 0.05. (B) Genes with significant differential expression. On the right side of the figures are the genes overexpressed in leaves and on the left side are the genes overexpressed in roots. The metatranscriptomic sample RmR3 was excluded from the analysis presented in the figure. Please refer to the Methodology section for further details.

Figure 6 Functional enrichment of coding DNA sequences (CDSs) from the endophytic microbiome of R. mangle, based on EggNOG database annotations.

(A) Functions enriched in leaf tissue. (B) Functions enriched in root tissue.

Metagenome + metatranscriptome: Role of microorganisms in plant-endophyte interactions

Annotation against the PLaBAse database showed that the microbial community of R. mangle is highly involved in xenobiotic metabolism, heavy metal resistance and salinity stress resistance. A general pattern of gene expression is observed with genes that are mostly expressed by endophytic microorganisms in roots and virtually none in leaves (Fig. 7). Within the group of abiotic stress resistance genes, the most abundant and active category in roots was tolerance to high temperature. For the heavy metal group, the most abundant and transcriptionally active categories were those related to resistance to arsenic, copper, nickel, and zinc. All categories related to phosphorus, sulfur, nitrogen and carbon metabolism were present and active. However, the metabolic category with the highest transcriptional activity was iron acquisition and homeostasis. Similarly, 17 categories related to the degradation of different xenobiotics were found, although their expression levels were not high compared to other categories. Finally, the category with the highest abundance and expression was the one related to salinity stress, presenting a pattern of gene expression opposite to that observed for the other categories, with greater activity in leaves than in roots (Fig. 7).

Figure 7 Heatmap of selected functions relevant to the R. mangle-endophyte interaction, based on PLaBAse database annotations.

DNA samples represent the abundance of coding DNA sequences (CDSs) assigned to functional categories in the metagenomes, while RNA samples reflect the expression levels of these categories in the metatranscriptomes. The metatranscriptomic sample RmR3 was excluded from the analysis presented in the figure. Please refer to the Methodology section for further details.

Discussion

In this study, a robust characterization of endophytic microorganisms from leaf and root tissues of R. mangle was performed. Shotgun and RNAseq sequencing methods were integrated to obtain the metagenome and metatranscriptome of each tissue. It was demonstrated that the endophytic microbiota of R. mangle is more diverse and active in the roots than in the leaves, the tissue type being an important factor in the composition of the microbiota of the internal compartments of the plant. It was also observed that the largest amount of CDS within the metagenome corresponded to functional categories related to cell maintenance, signaling and metabolism of macromolecules, which in turn were the categories most expressed in the metatranscriptome. On the other hand, it was found that endophytic microorganisms help R. mangle in various metabolic processes and confer resistance to salinity, heavy metals, biotic and abiotic stresses, and xenobiotic compounds. Taken together, the results suggest that endophytes of R. mangle roots and leaves complement the intrinsic morphophysiological adaptations of the plant to adapt to the extreme conditions of mangrove ecosystems.

Taxonomic composition of the endophytic microbiota of R. mangle

Identification of endophytic microorganisms associated with R. mangle revealed several genera of bacteria as the most abundant groups (Fig. 1D, Figs. S1–S6). The genus Bradyrhizobium has not been previously reported as an endophyte of R. mangle, but this group of bacteria can fix nitrogen and is a relevant endophyte to legume crops such as soybean and rice (Piromyou et al., 2015; Subramanian et al., 2015). Because bacteria of this genus are diazotrophic, they are considered to promote host plant growth, a function they may play in their interaction with R. mangle (Subramanian et al., 2015). Similarly, bacteria of the genus Streptomyces have been classified as plant growth promoting endophytes due to their production of a variety of metabolites against different phytopathogens (Vurukonda, Giovanardi & Stefani, 2018; Worsley et al., 2020). Many species of the genera Streptomyces and Nocardioides have been reported as endophytes of mangrove plants and as part of the soil of mangrove forests (Law et al., 2019). Several strains of the genus Pseudomonas promote host plant growth due to their ability to solubilize phosphates (Flores-Duarte et al., 2022). The genus Lactiplantibacillus has been reported to have antifungal activity but has not been described as an endophyte of R. mangle (Li et al., 2023). For the genus Mycobacterium, high-throughput biochemical profiling has revealed that some species have the potential to be endophytes, as they can metabolize carbon and nitrogen-containing substrates (Loukil et al., 2019).

Analyses of the leaf and root metagenomes revealed that the root tissue of R. mangle has a greater diversity in the composition of the endophytic microbial community (Fig. S10), and that tissue type may influence the structure of the endophytic microbiota (Fig. 2). Harrison & Griffin (2020) reported that tissue type is one of the factors determining the composition of the associated endophyte community within a single plant. This pattern has been observed previously in different investigations, where endophytes in leaf tissues are different from endophytes in root tissues (Cregger et al., 2018; Peng et al., 2022). For Rhizophora stylosa species, the diversity of endophytic fungi was found to decrease from roots to leaves due to intrinsic plant factors and environmental conditions (Purahong et al., 2019). Similarly, in R. mangle, metabarcoding revealed that bacterial endophytes differ between the rhizosphere and the different tissues of the plant (stems, leaves, flowers and fruits) (Scherer, Mason & Mast, 2022). The high diversity of endophytic microorganisms in root tissue can be attributed to the fact that this tissue is in direct contact with the soil, an area where microorganisms are abundant and highly active (Glick, 2020). On the other hand, endophyte diversity in leaves may be lower because environmental conditions such as temperature, UV radiation, and humidity are more variable than in roots, limiting the establishment and proliferation of microorganisms (Peng et al., 2022).

This difference could be attributed not only to the physicochemical heterogeneity of the root environment, but also to a higher metabolic exchange between the host and the endophytes in root tissues. Furthermore, the increased transcriptional activity observed in root-associated communities suggests that this compartment may harbor a higher proportion of metabolically active microorganisms that contribute to stress adaptation and nutrient acquisition. These findings underscore the functional specialization of endophytes depending on their plant tissue niche, beyond differences in environmental stability, a phenomenon also observed in other plant systems such as Solanum tuberosum L. (Petrushin, Filinova & Gutnik, 2024) and Arabidopsis thaliana (L.) Heynh (Bulgarelli et al., 2012), where root tissues have been shown to harbor specific microbial communities, structured by host filtering and niche specificity.

Beyond the general patterns observed between tissues, particular trends were identified that deserve attention. For example, leaf sample RmL2 showed a different endophytic microbial composition compared to the other replicates of the same tissue. This pattern could be related to genetic variation among individuals of R. mangle, considering that plant genotype has been widely recognized as a key factor shaping endophytic microbiota composition in other plants (Ji et al., 2023). In mangroves, Yao et al. (2020) reported that species identity has a marked effect on the associated microbial community, although the role of intraspecific genetic variation has not yet been assessed. Alternatively, specific microenvironmental conditions may also modulate microbiota structure at fine spatial scales (Zhuang et al., 2020). In addition, this study revealed a diverse set of low abundance taxa that, although not dominant, are part of the R. mangle microbiome (Fig. 1D). These rare taxa may play relevant ecological roles and represent an important reservoir of genetic diversity, highlighting the importance of exploring them in future research (Wainwright et al., 2023; Wang et al., 2022).

Functional potential of the R. mangle endophyte community: functional annotation of metagenomic coding sequences

Functional annotation of the leaf metagenome revealed abundant CDS associated with carbohydrate metabolism and transport (Fig. 3A & Fig. S8). This is consistent with the physiological role of leaves, where carbohydrates are abundant and essential to support non-photosynthetic tissues (Ainsworth & Bush, 2011) and are reabsorbed during senescence (Lin et al., 2010). In mangrove ecosystems, carbohydrates are a major carbon source, contributing up to 65.5% of tree carbon (Kristensen et al., 2008). Our results suggest that endophytes may exploit these resources and potentially contribute to carbohydrate metabolism and transport, as suggested in other systems (Saminathan et al., 2018). To explore whether this interaction extends to functional integration between host and microbiota, the hologenome theory of evolution provides a useful framework (Zilber-Rosenberg & Rosenberg, 2008), which proposes that host and microbiome together form a holobiont. Future studies could address this through combined meta-omics, plant genomics and transcriptomics. As for root tissues, CDS related to energy production and conversion were relevant (Fig. 3 & Fig. S8). This could be since root endophytes can solubilize elements such as phosphorus and zinc, processes that are important for the production and transfer of energy within the plant. It can also be explained by the presence of heavy metals in the sampling area (Gamboa-Garcia et al., 2020; Erazo Enríquez, 2022), since it has been recorded that some endophyte strains, as a response to heavy metal stress, produce indoleacetic acid, iron transporters and the enzyme ACC deaminase (Cui et al., 2024), which increases the expression of proteins involved in energy production. Now, in the metagenomes of leaf and root tissues, a category with high abundance of CDS was amino acid transport and metabolism (Fig. 3 & Fig. S8). In Helianthus annuus, it has been reported that endophytes play an important role in defense against infection through the production of amino acids and other compounds (Waqas et al., 2015). Considering the importance of amino acid metabolism in roots, their possible contribution to resistance to biotic stress could be considered. In addition, amino acids are part of root exudates, which are important for communication of the plant and endophytes with rhizosphere and soil microorganisms (Wang et al., 2021; Ali et al., 2024).

Functional annotation, differential expression and functional enrichment of the metatranscriptome

The results obtained by performing PCA on the metatranscriptome (Fig. 4) and differential expression analysis (Figs. 5A and 5B) indicate that active gene functions depend on the tissue in which the microorganisms are located. As mentioned previously, there are differences in intrinsic and microenvironmental conditions between leaves and roots (Purahong et al., 2019; Glick, 2020; Peng et al., 2022). These differences between tissues may affect not only the taxonomy, but also the gene expression of the microbiota, which is one of the possible explanations for the differences found in both PCA and differential expression analysis.

Among the genes overexpressed in leaves is glutamine synthetase (Fig. 5B), an enzyme that can be produced by both the plant and its endophytic microorganisms and is part of nitrogen metabolism, allowing the production of glutamine from glutamate and ammonia (Cipriano et al., 2021). Considering this result, it is suggested that leaf microorganisms may be involved in nitrogen metabolism. An example of this occurs in the species Camellia sinensis, where glutamine synthetase from leaf endophytes has been shown to contribute to the metabolism of glutamine and subsequent production of theanine, which serves as a nitrogen reserve molecule for the plant (Sun et al., 2019; Chang et al., 2023). Another case where endophytes have been shown to help modulate nitrogen metabolism in leaves occurs in sugarcane seedlings. In these seedlings, microorganisms express their glutamine synthetase and can even influence the production of this enzyme by the plant (Cipriano et al., 2021). A similar mechanism may occur in the leaves of R. mangle, where endophytic microorganisms contribute to the uptake and storage of nitrogen in the form of amino acids. This hypothesis is also supported by the fact that amino acid transport and metabolism was found to be the most abundant functional category in the functional enrichment analysis of overexpressed genes in leaves (Fig. 6A).

As for the roots, the results suggest that the microbiota of this tissue had an activity more focused on aerobic energy production than the microbiota of the leaves, since 39 genes were found with significant differential expression, among them the subunits of the enzymes cytochrome b, cytochrome c, and NADH dehydrogenase (Fig. 5B). In addition, functional enrichment against the EggNOG and KEGG databases revealed the oxidative phosphorylation pathway and energy production and conservation as enriched categories (Fig. 6B and Fig. S8). A possible explanation for this high energetic activity is that the root tissue sample was taken from the part of the root exposed to tidal action during low tide, when oxygen uptake through the root lenticels is facilitated (Inoue et al., 2024), suggesting that the aerobic energy-producing pathways of the microorganisms were active. In addition to the above-mentioned genes, acid protease transcripts were also found to be overexpressed, possibly due to the slightly acidic pH (pH 5.91) (Table S1), of the soil in which the R. mangle plants are found. Another overexpressed gene was that of cellobiohydrolase II, which is an enzyme widely studied at the biotechnological level because it allows cellulose degradation (Dai et al., 2021; Pramanik et al., 2021; Niu et al., 2024), having found this gene overexpressed may suggest the presence of a pathogenic endophytic fungus that degrades cellulose in the roots of R. mangle. Finally, the gene for the enzyme oxalate decarboxylase was also found to be overexpressed; oxalate is a molecule that can be produced by bacteria, fungi, and plants and participates in the phytoremediation of heavy metal contaminated soils (Prasad & Shivay, 2017; Graz, 2024). However, high levels of oxalate can be toxic to the plant (Li et al., 2022), so it can be hypothesized that the community of microorganisms that express oxalate decarboxylase will try to keep oxalate levels under control and that this molecule will not ultimately affect the plant.

Metagenome + metatranscriptome: role of microorganisms in plant-endophyte interactions

The comparison between CDS abundance obtained from the metagenome and gene expression obtained from the metatranscriptome showed that there is a higher transcriptional activity of coding sequences in root tissues compared to leaf tissues (Fig. 7). As mentioned above, these results may be due to differences in microenvironmental conditions between roots and leaves (Purahong et al., 2019; Glick, 2020; Peng et al., 2022). As a prominent functional activity, endophytes in the root tissue of R. mangle were found to contribute to the plant’s tolerance to high temperatures (Fig. 7). Although R. mangle has mechanisms to adapt to high temperatures (Nizam, Meera & Kumar, 2022), it has been shown that in mangroves of the equatorial region there is greater transcriptional activity associated with responses to high temperatures and UV radiation compared to subtropical mangroves (Bajay et al., 2018). The analyzed individuals are found in mangrove forests of the equatorial region, suggesting that abiotic stress induces a greater dependence on their endophytes, which is reflected in microbial transcriptional activity. The contribution of endophytic microorganisms to thermal tolerance has been reported in other species such as Glycine max and Serendipita indica (Bilal et al., 2020; Tyagi et al., 2022).

The endophytic microbiota of R. mangle also contributes to functions such as CO2 fixation, nitrogen acquisition, carbohydrate metabolism, phosphate solubilization, sulfur assimilation, and especially to homeostasis and iron acquisition (Fig. 7). These results are attributed to the fact that mangrove forests are ecosystems where a large amount of organic matter is produced and, in the plant-microorganism interaction, the functions are relevant for the flow of these nutrients within the ecosystem (Holguin, Vazquez & Bashan, 2001). Several studies have demonstrated the contribution of endophytes in these functional categories (Mei et al., 2021; Narayan et al., 2021; Chen et al., 2022; Qin et al., 2022). On the other hand, R. mangle endophytes are also involved in xenobiotic metabolism and heavy metal resistance (Fig. 7). This finding could be explained by the characteristics of the sampling area. The mangrove forest in Buenaventura Bay has a high degree of human intervention, which has caused contamination by plastics and heavy metals (Gamboa-Garcia et al., 2020; Erazo Enriquez, 2022; Bolivar-Anillo et al., 2023). The presence of these pollutants exposes mangrove plants and their endophytic microorganisms to constant abiotic stress, which is reflected in the high transcriptional expression of CDS associated with detoxification processes. The ability of endophytes to degrade xenobiotic compounds and heavy metals has been demonstrated in many works (Shahzad et al., 2019; Datta et al., 2020; Husna et al., 2021; Liu et al., 2021; Pandey, Kumar & Dey, 2024). Taken together, all these functions performed by endophytes are beneficial to R. mangle, providing macro- and micronutrients necessary for its development or protecting the plant against stress and damage that may be caused by environmental conditions in mangrove forests.

In addition to the above, an interesting function of the R. mangle endophyte interaction is salinity tolerance. Although the salinity recorded at the time of sampling was 18.5 practical salinity unit (PSU), (Table S2), Mondragon-Diaz, Molina & Duque (2022) have reported that this parameter presents a temporal variation depending on the rainy season, fluctuating between 15.8 and 22.2 PSU, which represents a constant salinity stress for the mangrove forest plant community and its associated microorganisms. This was evidenced by the abundance and high expression of CDS associated with the neutralization of osmotic stress in leaf tissues and the neutralization of salt stress in both leaves and roots (Fig. 7). R. mangle has several adaptations that give it tolerance to salinity, for example, its leaves have lenticels that allow the elimination of ions and specialized cells that retain water, ensuring an osmotic balance (Naskar, Mondal & Ankure, 2021; Bento et al., 2024). Also, the high levels of CDS expression associated with salinity tolerance suggest that endophytic microorganisms complement the intrinsic adaptations of the plant, providing a better adaptation to the intertidal zone. Vaishnav et al. (2019) reported that endophytic bacteria reduce osmotic pressure by producing antioxidants, osmolytes, and phytohormones. Subedi et al. (2022) showed that inoculation of R. mangle seedlings with endophytes increased salinity resistance (Subedi et al., 2022). The results of our metagenomic and metatranscriptomic analysis add to the available evidence supporting the involvement of endophytes, both leaf and root tissues, in salinity tolerance.

Conclusions

This study provides a complete characterization of the endophytic microbial community of red mangrove leaf and root tissues, integrating metagenomic and metatranscriptomic analyses. The results showed that the composition and gene expression of endophytes varied significantly among tissues and that roots were the compartment with the highest diversity and transcriptional activity. Endophytic microorganisms not only contribute to the metabolism of essential nutrients such as carbon, nitrogen, phosphorus and sulfur, but also play a crucial role in salinity tolerance and detoxification of heavy metals and xenobiotic compounds. These functions contribute to the adaptations of R. mangle and demonstrate the synergistic contribution of the microbiota in tolerance to the extreme conditions of an intervened mangrove ecosystem in Buenaventura Bay, Colombia. Overall, this work expands the knowledge of plant-microorganism interactions in extreme environments and positions the endophytic microbiota as a key component in the ecology of the red mangrove, as well as a reservoir of endophytic microorganisms with biotechnological potential. We propose as a next step to conduct a study with more sampling sites throughout the Colombian Pacific and at different seasons to determine how spatiotemporal and climatic changes affect the composition and gene expression of the endophytic microbiota and thus obtain a more complete characterization of the microorganisms associated with Rhizophora mangle.

Supplemental Information

Supplemental Information 1 Meteorological parameters recorded in the sampling area on June 14, 2023 according to Dimar (General Directorate of Maritime Ports of Colombia

Supplemental Information 2 Physicochemical parameters of soil and estuarine water from the sampling area

Supplemental Information 3 Number of reads before and after processing for DNA and RNA libraries

Supplemental Information 4 Gene expression counts from metatranscriptomic samples

Supplemental Information 5 Location of the sampling area

A). Buenaventura Bay in southwestern Colombia. B) Dagua River estuary into the Pacific Ocean. C) Sampling area inside the mangrove forest.

Supplemental Information 6 Fluctuations of the tide level in the sampling area on June 14, 2023

Data available at the Dimar (General Directorate of Maritime Ports of Colombia): https://www.dimar.mil.co/.

Supplemental Information 7 Experimental design for the collection and processing of R. mangle samples

Three replicates per tissue type (leaf: RmL1–3; root: RmR1–3). DNA and RNA were extracted from each individual sample for metagenomic and meta-transcriptomic analyses.

Supplemental Information 8 Taxonomic composition of the endophytic microorganisms associated with the RmL1 sample

Supplemental Information 9 Taxonomic composition of the endophytic microorganisms associated with the RmL2 sample

Supplemental Information 10 Taxonomic composition of the endophytic microorganisms associated with the RmL3 sample

Supplemental Information 11 Taxonomic composition of the endophytic microorganisms associated with the RmR1 sample

Supplemental Information 12 Taxonomic composition of the endophytic microorganisms associated with the RmR2 sample

Supplemental Information 13 Taxonomic composition of the endophytic microorganisms associated with the RmR3 sample

Supplemental Information 14 Alpha diversity indices of the endophytic microbiota of R. mangle

Supplemental Information 15 Dot plot of the functional annotation of the metagenomes of the R. mangle endophyte community against the KEGG database

A) Metagenome of leaf tissue. B) Metagenome of root tissue.

Supplemental Information 16 Functional fortification against the KEGG Database of Root Microorganisms

We thank Carlo Santiago Becerra Casierra for his support during the tissue processing in the laboratory.

Additional Information and Declarations

Competing Interests

Author Contributions

Field Study Permissions

Data Availability

The authors declare there are no competing interests.

Valentina Cárdenas-Hernández conceived and designed the experiments, performed the experiments, analyzed the data, prepared figures and/or tables, authored or reviewed drafts of the article, and approved the final draft.

Cesar Lemos-Lucumi conceived and designed the experiments, performed the experiments, analyzed the data, prepared figures and/or tables, authored or reviewed drafts of the article, and approved the final draft.

Nelson Toro-Perea conceived and designed the experiments, authored or reviewed drafts of the article, and approved the final draft.

The following information was supplied relating to field study approvals (i.e., approving body and any reference numbers):

The permit for the collection of leaf and root tissues was granted by the National Environmental Licensing Agency (ANLA) by Resolution 1070 of August 28, 2015.

The following information was supplied regarding data availability:

The processed reads of both DNA and RNA are available at NCBI: PRJNA1215819; SAMN46420086, SAMN46420087, SAMN46420088, SAMN46420089, SAMN46420090, SAMN46420091, SAMN46420092, SAMN46420093, SAMN46420094, SAMN46420095, SAMN46420096.

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
