# Peer review of "Uncovering tissue-specific endophytic microbiota composition and activity in Rhizophora mangle L.: a metagenomic and metatranscriptomic approach"

_PeerJ, doi:10.7717/peerj.19728_

## Round 0.1 · original submission · Major Revisions

- A clear experimental design should be provided. I suggest that in the Results section, the authors begin with the purpose of the study, followed by the results. It would be helpful if the selected methods are appropriate.

- Clear and well-labeled figures should be included.

- Importantly, it would be better if the authors could provide stronger evidence—either through additional experiments or by citing references that directly support their claims.

Reviewer 1 ·

Basic reporting

This study presents a metagenomic and metatranscriptomic analysis of endophytic microbiota in Rhizophora mangle. The dataset is valuable, and the study addresses an interesting topic. However, several aspects require further clarification and improvement, particularly in the methodological descriptions, as well as result presentation and interpretation.
Line 73-100: The introduction to metagenomics and metatranscriptomics is too general. Instead of broad definitions, focus on how these approaches address specific challenges in studying mangrove endophytes. Citing relevant examples or case studies on mangrove microbiomes would strengthen the rationale.

Experimental design

Line 125-127: Provide the rationale or cite relevant literature supporting the choice of this disinfection protocol. Additionally, clarify whether this protocol may influence microbial community composition.
Line 178-179: The statement "problems encountered in the construction of the library" is vague. Specify the issues faced.
Line 198-200: “In addition, through BUSCO v5.8.2, the metagenome was evaluated for contamination with host plant reads and, using CD-Hit v4.8.1,redundant sequences were removed from the metagenome.” Clearly describe the criteria used to identify host plant contamination and specify the CD-Hit parameters (e.g., sequence identity threshold) applied for filtering.

Validity of the findings

Line 267-271: There is a large discrepancy between the number of raw reads and retained reads. Only ~11.3% of DNA reads and ~1.3% of RNA reads were retained after processing. This low retention rate requires clarification.
Line 273-286: The taxonomic classification rates (8.19–24.7%) seem low. Were unclassified reads analyzed separately? If classification was based on a specific threshold (e.g., % identity), it should be stated.
Line 298-302: The statement “The leaf metagenome is significantly smaller than the root metagenome (6.5M vs. 915M bp)” lacks specificity. The size difference should be explicitly stated.
Lines 322-325: “Metagenome annotation against the KEGG database showed a similar pattern to the results obtained with EggNOG.” The term “similar pattern” is vague. Please clarify what aspects of the results were comparable.
Lines 341-352: “Differential expression analysis based on RefSeq revealed a total of 44 genes with significant differential expression, of which four were found to be overexpressed in leaves and 39 in roots.” The number of differentially expressed genes appears relatively small. Did you assess the reproducibility of these results using an alternative method, such as EdgeR?

·

Basic reporting

The introduction section lacks the review of recent works on the topic: metagenomics studies of the endophytic microbiome of mangrove plants. The following papers could help to improve the Intro and enrich the comparison in the Discussion:
Zhuang, W., Yu, X., Hu, R. et al. Diversity, function and assembly of mangrove root-associated microbial communities at a continuous fine-scale. npj Biofilms Microbiomes 6, 52 (2020). https://doi.org/10.1038/s41522-020-00164-6
Sui J, He X, Yi G, Zhou L, Liu S, Chen Q, Xiao X, Wu J. 2023. Diversity and structure of the root-associated bacterial microbiomes of four mangrove tree species, revealed by high-throughput sequencing. PeerJ 11:e16156 https://doi.org/10.7717/peerj.16156
Hui Yao, Xiang Sun, Chao He, Xing-Chun Li, Liang-Dong Guo, Host identity is more important in structuring bacterial epiphytes than endophytes in a tropical mangrove forest, FEMS Microbiology Ecology, Volume 96, Issue 4, April 2020, fiaa038, https://doi.org/10.1093/femsec/fiaa038
Figure 1 and 2 has poor quality, captions are hard to read.

Experimental design

L198: as far as I knom BUSCO cannot detect the contamination, this tool assesses the completeness of the genome, its abilities to evaluate the metagenome is limited so far.
Figure 1 colors would be contrast.
Figure 5A is unclear and hard to understand the caption meanings, please revise the figure layout.
The sample RmR3 is missing at Figure 4.

Validity of the findings

At Figure 1D genus “ab. < 4%” takes the most part of the microbial community?
How authors interpret the difference of community structure of the sample RmL2 from other samples?

Additional comments

Minor text issues:
L665: the alphabet order has to be fixed

·

Basic reporting

The aim of the study is to integrate metagenomics and metatranscriptomics to characterize the endophytic microbiota associated with the leaf and root tissues of Rhizophora mangle. This research intended to fill the gap of knowledge on endophytes in mangrove forests and enhance our understanding of plant-endophyte interactions within challenging intertidal environments.

The quality and clarity of fugyres 2, 3 and 5 would be improved.
Why we see only 5 objects at figure 4 while there are 6 samples in the study?

Experimental design

pass

Validity of the findings

L423-430 The authors found out as a result of their research that the endophytic leaves microorganisms have more genes responsible for the metabolism and transport of carbohydrates than those in the root tissue. The authors explain this by the fact that these microorganism genes help the transport and metabolism of carbohydrates in the plant itself. This statement can be interpreted as the fact that the microorganism genes are involved in the functioning of the plant organism, that is, they act in conjunction with the plant genome. The authors refer to the works of Saminathan et al., 2018; Fadiji et al., 2021.

The work of Saminathan et al. does indeed contain indirect evidence of such a phenomenon, but I did not find convincing evidence in the work of Fadiji et al. Meanwhile, such a statement requires more detailed argumentation. It would be interesting to know the authors’ opinion on the hologenome theory (here is an example of an article about this: Zilber-Rosenberg I., Rosenberg E. Role of microorganisms in the evolution of animals and plants: the hologenome theory of evolution // FEMS Microbiol Rev. 2008. V. 32(5). P. 723–35.) The authors can also read the following article: Bacterial genome evolution in superspecies systems: an approach to the reconstruction of symbiogenesis processes Provorov N.A., Tikhonovich I.A. Russian Journal of Genetics. 2015. Т. 51. № 4. С. 377-385.

I think that before putting forward such a version, the authors should first propose a more obvious version, which is that there are many genes for carbohydrate transport and metabolism in endophytic leaf microorganisms, since it is in the leaves that there is an excess of carbohydrates.

Additional comments

The references list should be checked carefully to contain DOI of full description.

---

## Round 0.2 · Minor Revisions

The manuscript has significantly improved. However, there are still some minor comments for the authors to address, particularly regarding the data sources, typographical errors, and the clarity of the discussion.

Reviewer 1 ·

Basic reporting

Reporting of the work is clear.

Experimental design

Description on experimental design reads ok in the revised version.

Validity of the findings

The results are presented appropriately.

Additional comments

The authors have responded to all the reviewer’s queries.

·

Basic reporting

L114: the BioProject contains 23 samples. Please describe the details about samples in "Experimental design and sample collection" subsection

L171 says "The NovaSeq6000 platform (Illumina) was then used for paired-end sequencing of 150 bp fragments, generating 20 GB of information for each sample." but SRA records contain much less data (max 6.9G bases)

L389: "It was demonstrated that the endophytic microbiota of R. mangle is more diverse and active in the roots than in the leaves" that is well-known fact because of more stable conditions in leaves niche (we also stress this point in our review on potato microbiome https://doi.org/10.3390/ijms25020750). What else could be said about difference of rhizosphere and endosphere microbiome composition?

Experimental design

pass

Validity of the findings

pass

Additional comments

Minor text comments:
L405: "endophyte in legume crops" to "endophyte to legume crops"

L439: "R. mangle" italic font

L566: "PSU" please explain the abbreviation

·

Basic reporting

The authors have done a lot of work, the results of which are very useful for understanding the composition and function of the endosphere and rhizosphere of Rhizophora mangle L. The authors have taken into account all the comments, and the edits made are quite satisfactory. I hope that the authors will continue their research towards studying the results of the coevolution of the plant and its symbionts.

Experimental design

pass

Validity of the findings

pass

---

## Round 0.3 · accepted · Accept

The manuscript has significantly improved, and its quality meets the scientific standards for publication.